# Antimicrobial Resistance in Bacteria Isolated from Exotic Pets: The Situation in the Iberian Peninsula

**DOI:** 10.3390/ani12151912

**Published:** 2022-07-27

**Authors:** Eleonora Muñoz-Ibarra, Rafael A. Molina-López, Inma Durán, Biel Garcias, Marga Martín, Laila Darwich

**Affiliations:** 1Department Sanitat i Anatomia Animals, Veterinary Faculty, Universitat Autònoma de Barcelona (UAB), 08193 Cerdanyola del Vallès, Spain; munozeleonora@gmail.com (E.M.-I.); biel.garcias@uab.cat (B.G.); marga.martin@uab.cat (M.M.); 2Catalan Wildlife Service, Centre de Fauna Salvatge de Torreferrussa, 08130 Santa Perpètua de Mogoda, Spain; rafael.molina@gencat.cat; 3Departamento de Veterinaria de Laboratorio Echevarne, 08037 Barcelona, Spain; idduran@laboratorioechevarne.com

**Keywords:** exotic pets, antimicrobial resistance, birds, mammals, reptiles, bacteria

## Abstract

**Simple Summary:**

Antimicrobial resistance in exotic pets has not been widely studied. The close contact of this type of animal with the human population increases the risk of untreatable bacterial infections, which represent a veterinary and human public health challenge. We analyze the database of microbiological diagnoses and the bacterial susceptibility to antimicrobials in exotic pets from the Iberian Peninsula. We found that the most prevalent bacteria in birds and mammals were *Staphylococcus* spp., while in reptiles, they were the *Pseudomonas* spp. In addition, *Pseudomonas* showed the highest levels of resistance among the three animal groups, and on the other hand, the multidrug resistance level was significant in Enterobacterales. Most of the bacteria we found have zoonotic importance. The prevalent bacteria are resistant to antimicrobials that have been described as critical for human use, implying that the threat of antimicrobial resistance extends not only to domestic and companion animals but also to humans due to the potential transmission of resistant genes. Once seen from the lens of the One-Health paradigm, these findings are concerning, as they highlight the risk of spreading antibiotic-resistant genes between different individuals and their environments. In order to prevent antibiotic resistance, we encourage the development of joint work between animal and human health specialists.

**Abstract:**

Literature related to antimicrobial resistant (AMR) bacteria in exotic pets is minimal, being essential to report objective data on this topic, which represents a therapeutic challenge for veterinary medicine and public health. Between 2016 and 2020, laboratory records of 3156 exotic pet specimens’ microbiological diagnoses and antibiotic susceptibility testing (AST) results were examined. The samples were classified into three animal classes: birds (*n* = 412), mammalia (*n* = 2399), and reptilian (*n* = 345). The most prevalent bacteria in birds and mammals were *Staphylococcus* spp. (15% and 16%), while in reptiles they were *Pseudomonas* spp. (23%). Pseudomonas was the genus with the highest levels of AMR in all animal groups, followed by *Enterococcus* spp. By contrast, Gram-positive cocci and *Pasteurella* spp. were the most sensitive bacteria. Moreover, in reptiles, *Stenotrophomonas* spp., *Morganella* spp., and *Acinetobacter* spp. presented high levels of AMR. Multidrug-resistant (MDR) bacteria were isolates from reptiles (21%), birds (17%), and mammals (15%). The Enterobacterales had the highest MDR levels: *S. marcescens* (94.4%), *C. freundii* (50%), *M. morganii* (47.4%), *K. pneumoniae* (46.6%), *E. cloacae* (44%), and *E. coli* (38.3%). The prevalence of MDR *P. aeruginosa* strains was 8%, detecting one isolate with an XDR profile. Regarding antimicrobial use, many antibiotics described as critically important for human use had significant AMR prevalence in bacteria isolated from exotic pets. Under the One-Health approach, these results are alarming and of public health concern since potential transmission of AMR bacteria and genes can occur from exotic pets to their owners in both senses. For this reason, the collaboration between veterinarians and public health professionals is crucial.

## 1. Introduction

There is increasing worry about the development of antibiotic resistance due to the numerous instances of bacterial infections that have affected humans and animals and may be fatal or exceedingly challenging to treat. Several studies have shown that the shift of resistant bacteria or genes through pets to humans is highly probable since pets play a role as a potential reservoir of AMR [1,2,3,4,5,6,7,8]. Although it is important to note that spillover transmission from people to animals is feasible, research on this situation is limited. Some worldwide studies have evidenced the transmission of diseases from exotic pets to their owners, handlers, or other animal species [2,3,9] but most of the monitoring programs are focused on antimicrobial use in livestock animals, and recommendations and policies that control or rule antibiotics stewardship in exotic pets are lacking [10].

In Spain, there are few studies conducted on exotic pets showing the potential risk of human infections by zoonotic bacteria such as *Salmonella* spp. in turtles [11,12] or respiratory infectious diseases in guinea pigs and chinchillas [13]. Although these studies have shown concern for investigating bacterial infections in exotic pets, there is a lack of published literature about the antimicrobial sensitivity profiles of bacterial infections in exotic pets against common antibiotics used in veterinary and human medicine. This information is critical since vet clinicians have no guidelines for selecting the best empirical antimicrobial treatment in terms of the risk of selection and dissemination of antibiotic resistance in exotic pets and its repercussion on the health of people and other domestic animals.

The present study analyses the clinical microbiological data on exotic pets collected between 2016 and 2020 by a large Diagnostic Laboratory in Spain to determine the most prevalent bacterial infections and AMR profiles among different exotic pet classes. The final objective is to provide evidence of the risk of zoonotic transmission of AMR bacteria between exotic pets and their owners in order to optimize treatment and preventive measures in veterinary practices.

## 2. Materials and Methods

### 2.1. Database Source and Management

The database used for this study comprises recorded data of the microbiological outcomes from clinical samples that contain information about clinical cases submitted by veterinary clinics throughout Spain and Portugal (Figure 1). The records were provided by the Veterinary Medicine Department of a large private laboratory of diagnosis in Barcelona (Spain). The lab has held the quality management system certificate ISO-9001 since 1998 and the accreditation from ENAC (National Accreditation Entity) according to criteria included in the ISO/IEC 17025 Standard defined in the Technical Annexes 511/LE1947 for Pharmaceutical Toxicology and Microbiology Testing.

The first step was to preclean the database, excluding the results without animal species information, type of sample, or antibiogram results. Likewise, the data were screened for samples collected in duplicate and those that were found to be repeated. The following variables were extracted from the records: animal species, type/origin of the sample, geographic origin of the specimen, bacterial identification, and antimicrobial susceptibility testing. Then, the animals were segregated following the categorization of animals depending on their purpose, behavior, or habitat, as mentioned in the Legislative Decree 2/15 April 2008 (Anon, 2011).

### 2.2. Study Design and Animal Samples

A total of 3986 microbiological analyses from clinical specimens taken between 2016 and 2020 comprised the final sample size. The sample was classified according to the exotic pet class as follows: birds (*n* = 591 samples), mammals (*n* = 3003), reptiles (*n* = 392).

In addition, data were grouped into the following categories: respiratory (nasal and bronchioalveolar samples); skin/external mucous (skin, turtle shell, ocular, otic, and other mucous); digestive (liver, gastrointestinal tract, and cloaca); urinary (urine and cloaca); musculoskeletal (bones, joints, and muscles); lymphoreticular (lymph nodes, coelomic cavity, and glands); embryo/egg/eggshell (fetus, yolk sacs, and amniotic liquid); reproductive (vaginal and penis secretions, and gonads); and circulatory (pericardium, heart, or blood).

### 2.3. Microbiological Diagnosis Techniques and Antimicrobial Susceptibility Testing

Microbiological identification was performed using the MALDI-TOF mass spectrometer or the API R ID system (bioMérieux, Madrid, Spain), as previously described by Darwich and others [1] and Li and others [6]. All Gram-positive bacterial isolates were performed by the antimicrobial susceptibility test (AST) using the standard disk diffusion method according to Performance Standards for Antimicrobial Susceptibility Testing for bacteria isolated from animals (M31-A3, CLSI VET01, 2008) and humans (M100-S24, CLSI, 2016) for monitoring resistant microorganisms as a potential risk to public health. The panel included 45 antimicrobials corresponding to 8 classes or categories and 7 single drug classes and their respective disc concentration: β-lactams (amoxicillin (AMO 30 µg), AMO + clavulanic acid (AMC/30 µg), oxacillin (OXA/1 µg), cefoxitin (CXI/30 µg), penicillin (PEN/10 U), piperacillin (PIP/110 µg), piperacillin/tazobactam (PIT/110 µg), ampicillin (AMP/10 µg), cephalexin (CLE/30 µg), cephalothin (CET/30 µg), cefazolin (CZO/30 µg), cefuroxime (CUR/30 µg), ceftazidime (CTZ/30 µg), cefotaxime (CTA/30 µg), cefovecin (CVN/30 µg), and cefepime (CEP/30 µg)), imipenem (IMI/10 µg), meropenem (MER/10 µg), and aztreonam (AZT/30 µg); fluoroquinolones (ciprofloxacin (CIP/5 µg), enrofloxacin (ENR/5 µg), nalidixic acid (NAL/30 µg)); aminoglycosides (amikacin (AMI/30 µg), gentamicin (GEN/10 µg), tobramycin (TOB/10 µg), neomycin (NEO/30 µg), kanamycin (KAN/30 µg)); macrolides (azithromycin (AZI/15 µg), erythromycin (ERY/5 µg)); tetracyclines (tetracycline (TET/30 µg), doxycycline (DOX/30 µg)); lincosamides (clindamycin (CLI/2 µg), lincomycin (LCM/2 µg)); polymyxins (polymyxin B (PMB/300 µg)); trimethoprim/sulfamethoxazole (TRS/25 µg); phenicols (chloramphenicol (CHL/10 µg), florfenicol (FLO/30 µg)); fosfomycin (FOS/50 µg); mupirocin (MUP/200 µg); metronidazole (MET/5 µg); glycopeptides (vancomycin (VAN/30 µg)); fusidic acid (FUS/10 µg); nitrofurantoin (NIT/300 µg), and rifampicin (RIF/5 µg).

For Gram-negative bacteria, NM44 MicroScan (Beckman Coulter, Villepinte, France) system testing was performed for all the antimicrobials except for those authorized for veterinary uses that are not included in the automatic scan panels (enrofloxacin, pradofloxacin, marbofloxacin, doxycycline, cephalexin, and cefovecin) [6]. Additionally, quality control for the AST was performed using internal controls in each automatic panel of NM44 MicroScan (Beckman Coulter, Villepinte, France). In the case of manual antibiograms, McFarland standards were used as a reference, previously confirmed by a Densicheck (bioMérieux, Madrid, Spain).

Isolates were classified as susceptible, intermediate, or resistant according to the results obtained from the lab records. In addition, multidrug resistance (MDR) was defined as resistance to at least one agent in ≥3 antimicrobial categories; extensive drug resistance (XDR) as resistance to all but two of the tested antimicrobial categories; and finally, pan-drug (PDR) as resistance to all the categories tested [14]. These classifications and definitions were also used in the MDR analysis.

### 2.4. Statistical Analysis

The statistical analysis was performed using R version 4.1.0 (R v 4.1.0 (R Core Team, Vienna, Austria), 2021) [15], applying the novel AMR package [16]. The statistical unit is bacterial culture results, which were considered individually for the analysis. Bacterial names were first manually and then automatically classified by the AMR package, the taxonomy of microorganisms was extracted from the Catalogue of Life database and the List of Prokaryotic names with Standing in Nomenclature; the interpretation of disk diffusion values were based on the CLSI and EUCAST guidelines available between 2011 and 2020, which are included in this package as well as antibiotic and AMR analysis.

Inconsistencies in the classification were manually reviewed. For statistical assessments, the package performed an accurate analysis avoiding the false susceptibility or bias susceptibility frequency of microorganisms, reporting the AST, regardless of the intrinsic resistance or taking out the repeated cases. The pack of functions worked with Chi-square (*X^2^*) test function to compare the animals’ classes, bacteria species, and the AMR frequencies; in this regard, the statistical significance was considered when *p* < 0.05.

## 3. Results

### 3.1. Microbiological Results in Exotic Pets

The results of the microbiological cultures were performed once the database was cleaned from duplicated results and the fungus cultures were removed. Consequently, the final data of the microbiological testing were 3156 samples that presented a positive diagnosis confirmation of a pure or majoritarian bacterial infection (Table 1).

From the microbiological results, birds (*n* = 412) represented 13%, mammals (*n* = 2399) represented 76%, and reptiles (*n* = 345) 11% of the total sample (Table 2). Within the birds, psittacines represented 85.2% of the avian class, with a particular frequency of *Agapornis* spp. (lovebirds) with 20%. The mammal class was principally represented by lagomorphs (73.2%) and rodents (22%), with *Oryctolagus cuniculus* (European rabbit) and the *Cavia porcellus* (guinea pig) being the principal species with 56% and 11% of the mammals, respectively. Finally, *Trachemys scripta* (red-eared slider turtle) represented 25% among reptiles, followed by 14% of *Python regius* (royal python).

According to the sample’s origin, skin/external mucous samples were the most frequent ones, followed by the respiratory specimens. Digestive problems were more common in birds and reptiles, while urinary infections were most important in mammals. Detailed information about the sample origin is depicted in the Appendix A Figure A1.

The most prevalent bacterial genus was *Staphylococcus* spp. (14.9%) and *Pseudomonas* spp. (14.3%), followed by *Streptococcus* spp. (9%) and Enterobacteria (*Klebsiella* spp. 6.6%, *Enterobacter* spp. 6.4%, and *Escherichia* spp. 6.3%) (Table 3). Within the birds and mammals groups, *Staphylococcus* spp. was the predominant agent, representing 16% of the cases. In reptiles, *Pseudomonas* spp. was the most frequent one representing 23% of the cases (Table 3).

In birds and mammals groups, a similar distribution of the bacterial agents was found for the different system categories (Figure 2a,b). In general, *Pseudomonas* spp. was the most important agent found in most categories, except in the urinary and musculoskeletal systems, where the most prevalent bacteria were *E. coli* and *Staphylococcus* spp., and the lymphoreticular system, with *Enterococcus* spp. as the principal agent. *Staphylococcus* spp. and Enterobacteria such as *E. coli*, *Klebsiella* spp., and *Enterobacter* spp. were also frequently isolated from digestive, respiratory, and skin/external mucous systems.

In reptiles, *Pseudomonas* spp. was also the most common agent isolated from almost all the system categories, except for the lymphoreticular system, where the principal bacteria were *Aeromonas* spp. and *Citrobacter* spp. These two former bacterial species were also found in other systems (digestive, respiratory, and skin/external mucous systems). In contrast to birds and mammals, reptiles presented a high prevalence of *Enterococcus* spp. in urinary infections (Figure 2c).

### 3.2. Prevalence of Antimicrobial-Resistant Bacteria

The frequencies of AMR among birds, mammals, and reptiles’ classes for the most frequent bacterial genus are in the Appendix A and shown in Figure 2, Figure 3 and Figure A1. Overall, Pseudomonas was the genus with the highest levels of AMR, whereas Gram-positive cocci (*Staphylococcus*, *Streptococcus*, or *Aeromonas* spp.) and *Pasteurella* spp. were the most sensitive bacteria in all animal classes. Moreover, in birds and mammals, *Enterobacter* spp. and *Klebsiella* spp. also presented high levels of AMR to a large number of antimicrobials, followed by *Enterococcus* spp. In reptiles, *Stenotrophomonas* presented the highest resistance levels in a global assessment, followed by other highly resistant genus such as *Morganella* spp. and *Acinetobacter* spp. Finally, AMR in E. coli was intermediate, normally lower levels than other members of the Enterobacterales order such as *Enterobacter*, *Klebsiella*, *Citrobacter*, or *Serratia* spp.

### 3.3. Frequency of Multidrug Resistance Profiles

The study of antimicrobial susceptibility patterns showed that 97% of the bacteria were resistant to at least one agent of the total antimicrobial categories: 99.5% in birds, 96% in mammals, and 98.5% in reptiles. Moreover, 16% of the microorganisms presented a MDR pattern. No cases of PDR were observed. The percentage of strains with higher MDR levels found in reptiles was 21%, in birds it was 17%, and in mammals 15%. Enterobacterales presented the most significant frequencies of MDR, remarking the prevalence of *S. marcescens* (94.4%), *C. freundii* (50%), *M. morganii* (47.4%), *K. pneumoniae* (46.6%), *E. cloacae* (44%), and *E. coli* (38.3%). Gram-positive cocci showed MDR frequencies of 24.6% for *S. aureus* and 15.4% for *E. faecalis*. Regarding *P. aeruginosa*, 8% of the isolates presented MDR patterns and one of the isolates showed an XDR profile.

In birds, most bacterial species showed high resistance to macrolides, fusidic acid, lincosamides, nitroimidazoles, and aminopenicillins. Additionally, *E. cloacae*, *S. marcescens*, and *P. aeruginosa* were highly resistant to 1st and 2nd generation cephalosporins, amoxicillin, clavulanic acid, and nitrofurans. For *P. aeruginosa*, this resistant pattern was also expanded to tetracyclines, chloramphenicol, and trimethoprim-sulfonamides (Figure 3).

In mammals, all bacterial species, with the exception of *S. aureus*, were highly resistant to fusidic acid, aminopenicillins (except for *E. coli*), nitroimidazoles and macrolides (except for *C. freundii*), and lincosamides (except for *P. aeruginosa*). Moreover, *E. cloacae*, *S. marcescens*, and *P. aeruginosa* were also resistant to 1st and 2nd generation cephalosporins and aminopenicillins β-lactamase inhibitors, and *S. marcescens* and *P. aeruginosa* expanded their resistance to 4th generation fluoroquinolones, chloramphenicol, and tetracyclines (Figure 3).

In the reptiles, all bacteria were highly resistant to fusidic acid, lincosamides, and macrolides. Furthermore, except for *E. coli*, all the bacteria showed high resistance to 1st and 2nd generation cephalosporins. In addition, most enterobacteria and *P. aeruginosa* demonstrated to have high resistance to penicillins, aminopenicillins, aminopenicillins β-lactamase inhibitors, and tetracyclines. *P. aeruginosa* also showed extended AMR to chloramphenicol and trimethoprim/sulfonamides (Figure 3).

## 4. Discussion

This work reports novel data about AMR in bacteria isolated from exotic pets in the Iberian Peninsula. These results provide helpful information for veterinarian clinicians and the scientific community since AMR in exotic pets can represent both a serious animal and public health concern. In this study, the most prevalent bacteria in birds and mammals were *Staphylococcus* spp. (16%), while in reptiles they were *Pseudomonas* spp. (23%). The AST results showed Pseudomonas presented the highest AMR levels in birds and mammals, detecting one isolate with an XDR profile, while in reptiles, it was found to be *Stenotrophomonas* spp., *Morganella* spp., and *Acinetobacter* spp. Moreover, the highest levels of MDR were observed in Enterobacterales: *S. marcescens* (94.4%), *C. freundii* (50%), *M. morganii* (47.4%), *K. pneumoniae* (46.6%), *E. cloacae* (44%), and *E. coli* (38.3%). Regarding antimicrobial use, many antibiotics described as critically important for human use had significant AMR prevalence in bacteria isolated from exotic pets. Thus, it is essential to deal with the burden of AMR, select the appropriate therapy, and research effective antibiotic stewardship in these animals.

According to the antibiotic categorization for prudent and responsible use in animals, made by AMEG [17], the antibiotic classes that clinical veterinarians could use to combat specific bacterial infections are those included in the C -caution- or D -prudence- categories authorized for vet use as a first-line treatment options. Among these categories can be found the 1st generation of cephalosporins, aminoglycosides, chloramphenicol, penicillin/β-lactam inhibitor, tetracyclines, and trimethoprim/sulfonamides. On the other hand, the antibiotics included in the A -avoid- or B -restrict- categories are the 3rd and 4th generation of cephalosporins; and 2nd and 3rd generation of fluoroquinolones, polymyxins, carbapenems, which are not authorized in veterinary medicine, cannot be used unless there is no other option, or shall always be used based on AST results.

The results of AMR bacteria obtained in exotic pets agree with data reported on dogs and cats in Spain [1,6]. The AST demonstrated that *Pseudomonas* spp. and *Enterococcus* spp. presented the highest levels of AMR in both dogs and cats and in exotic pets. Moreover, within the Enterobacterales, *E. coli* showed low levels of AMR compared with *K. pneumoniae*, *E. cloacae*, *S. marcescens*, *C. freundii*, and *M. morganii*. Even so, in these common pets, *Pasteurella* isolates were highly sensitive to all antimicrobials tested. In exotic pets, the most sensitive bacteria were also *Pasteurella* spp., followed by *Staphylococcus* spp. and *Streptococcus* spp. Furthermore, many bacterial isolates observed in this study have been described as important zoonotic bacteria in human infectious diseases [18,19,20]. On the other hand, the prevalence of MDR profile in overall exotic pet cases was 16%, higher than that reported in urinary tract infections (UTI) in dogs and cats (8%) [1]. Analyzing individual cases, *E. coli*, *K. pneumoniae*, and *S. aureus* from exotic pets presented lower resistance patterns than the equivalent isolates in dogs and cats, whereas *E. cloacae*, *S. marcescens*, and *P. aeruginosa* presented higher resistance profiles [1].

The results of AST in exotic pets reveal that *P. aeruginosa* was the bacteria with the largest frequency of AMR in all animal categories, followed by *E. cloacae* and *S. marcescens* in mammals, which commonly require the use of antimicrobials as a last resort for human medicine. *P. aeruginosa* is an opportunistic pathogen that causes nosocomial infections, pulmonary infections in cystic fibrosis patients, disseminated infections in immunocompromised humans [21], and severe infections in domestic and companion animals [1,6]. *P. aeruginosa* is also one of the most critical AMR priority pathogens, according to the World Health Organization [22]. There is an increase in the circulation of strains resistant to several antibiotic classes, including carbapenems [23]. Carbapenems are last-resort antibiotics used to treat serious infections by MDR bacteria, including *P. aeruginosa*. In our study, some *P. aeruginosa* avian strains presented an XDR profile, intrinsically resistant to beta-lactams and combinations with b-lactamase inhibitors, chloramphenicol, erythromycin, and trimethoprim/sulfamethoxazole. Furthermore, *Pseudomonas* spp., principally in birds and mammals, were also resistant to the aminoglycosides (around 60%), carbapenems (50% approx.), and second-generation fluoroquinolones (50% approx.). By contrast, approximately 70% of the isolates in mammals were susceptible to lincomycin, polymyxins, and third-generation fluoroquinolones. In comparison to pseudomonal infections in canines and felines of the Iberian Peninsula, exotic pets presented higher frequencies of AMR for enrofloxacin and aminoglycosides than dogs and cats [6]. These results represent a serious health concern since the circulation of these MDR *P. aeruginosa* strains is increasing among exotic pets, highlighting its global spread. Another related Gram-negative oxidase-positive bacterium often seen as a co-organism along with *P. aeruginosa* is *Stenotrophomonas* spp., detected in this study as MDR strains in the respiratory tract of reptiles. Stenotrophomonas can cause opportunistic infections in humans owing to biofilm formation and antibiotic resistance. Occasionally, this bacterium has been involved in sepsis and severe lung infections in immunocompromised patients [24].

Regarding Enterobacterales, *Morganella morganii* was the enterobacteria with the largest resistance frequencies in birds and reptiles. In mammals, *E. cloacae*, *S. marcescens*, and *K. pneumoniae* presented MDR profiles, making them difficult to combat with conventional antimicrobials for veterinary use. *M. morganii* causes urinary infection and, to a lesser extent, other gynecological-related infections or contaminating surgical wounds. Occasionally, it has been related to septic arthritis, especially in elderly patients with long-standing diseases [25]. On the other hand, in this study, *K. pneumoniae* only presented low resistance to carbapenems, which are reserved for critical use in human medicine; considering other antimicrobials authorized for vet medicine, the best options were aminoglycosides and chloramphenicol, with around 50% of the isolates presenting resistance to these drugs. An increasing problem for doctors globally is *K. pneumoniae*, as it is a MDR infection and is considered a zoonotic agent of great relevance to both animal and human health [26,27]. Studies on bacterial resistance in animals, food, and the environment have revealed that multidrug-resistant and extended-spectrum β-lactamase (ESBL) strains are also resistant to carbapenems and β-lactam group drugs. For instance, samples of soft tissue, respiratory tract, genital tract, urinary tract infections, wounds, and feces from domestic and companion animals have contained MDR and ESBL carbapenemase-producing strains of this bacteria [6,27,28]. As a consequence, it is imperative to improve the monitoring of MDR bacteria and prevent their evolution, such as *K. pneumoniae* and *P. aeruginosa* in animals, i.e., minimizing antibiotic exposure in veterinary by the use of microbiological diagnostic techniques for effective treatments. It is equally critical to monitor for and adopt hygiene methods in order to reduce the spread of pathogens in practice.

From a zoonotic point of view, the high levels of AMR to critically important antibiotics in human medicine found in exotic pets are of great concern since potential transmission of resistant genes from pets to humans or other animals can occur, considering that the predominant bacteria in this study are among the six pathogens directly attributed to human deaths due to AMR; likewise, they are significantly present in the urban microbiome with resistant genes [29,30]. Regarding antimicrobial use, many of the antibiotics described by the WHO as critically important for human use had significant AMR prevalence in bacteria isolated from exotic pets. Furthermore, under the AWaRe classification by the WHO [31], the principal bacteria found in this study present clear resistant patterns to different antibiotics used in human medicine (See Appendix A Table A1). It is necessary to highlight the susceptible bacterial patterns recommended by this study and its comparison with those allowed for animal use; the therapies shaped from this interpretation should be in contrast with the individual animal case and, as much as possible, with AST. Additionally, performing the susceptibility test with the antibiotics for veterinary medicine might support decision-making by veterinarians, prudent antibiotic use, and the reduction in the use of antimicrobials sensitive to human medicine.

From the One Health standpoint, it is critical that veterinarians and physicians work together to optimize, rationalize, and prudently use antimicrobial therapies in domestic, companion, and exotic animals and humans since most bacterial pathogens and their resistance mechanisms can be shared between animals and humans. Examples include adopting interrelated education on zoonotic infections and ownership implications in health, as well as exchanging data sources, experiences, and discussing clinical cases.

The results of this study provide objective data on the microbiological results (AMR bacteria and AST profiles) obtained in exotic pets of the Iberian Peninsula. On the one hand, these data can be useful for vet clinicians to apply empirical therapy in exceptional situations where the severity of the disease requires immediate antimicrobial treatment, with no time for AST analysis. On the other hand, this situation of AMR bacteria found in exotic pets to critically important antibiotics in human medicine is significant and a public health concern since potential transmission of resistant genes from exotic pets to humans or other animals can occur in both senses.

These findings are concerning in terms of a One-Health approach since they indicate the likelihood of resistance genes spreading through animals, the environment, and humans. For this reason, the collaboration between veterinary and public health professionals to combat AMR is essential.

## Figures and Tables

**Figure 1 animals-12-01912-f001:**
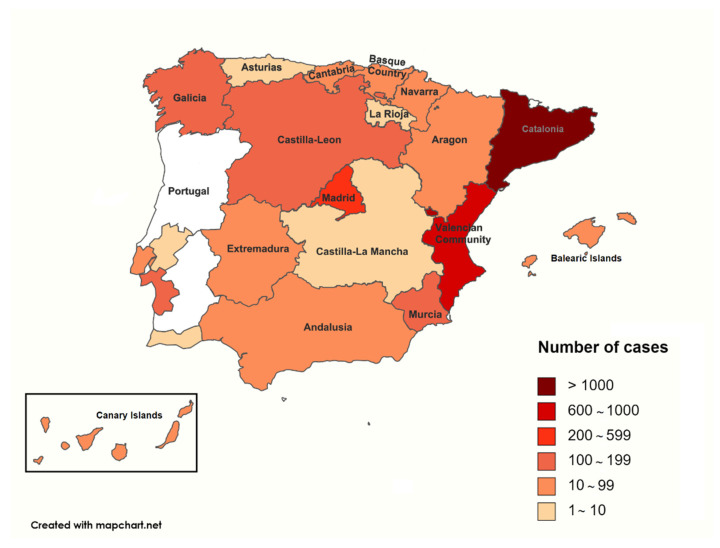
Distribution of clinical samples by geographical origin in Spain and Portugal analyzed between 2016 and 2020.

**Figure 2 animals-12-01912-f002:**
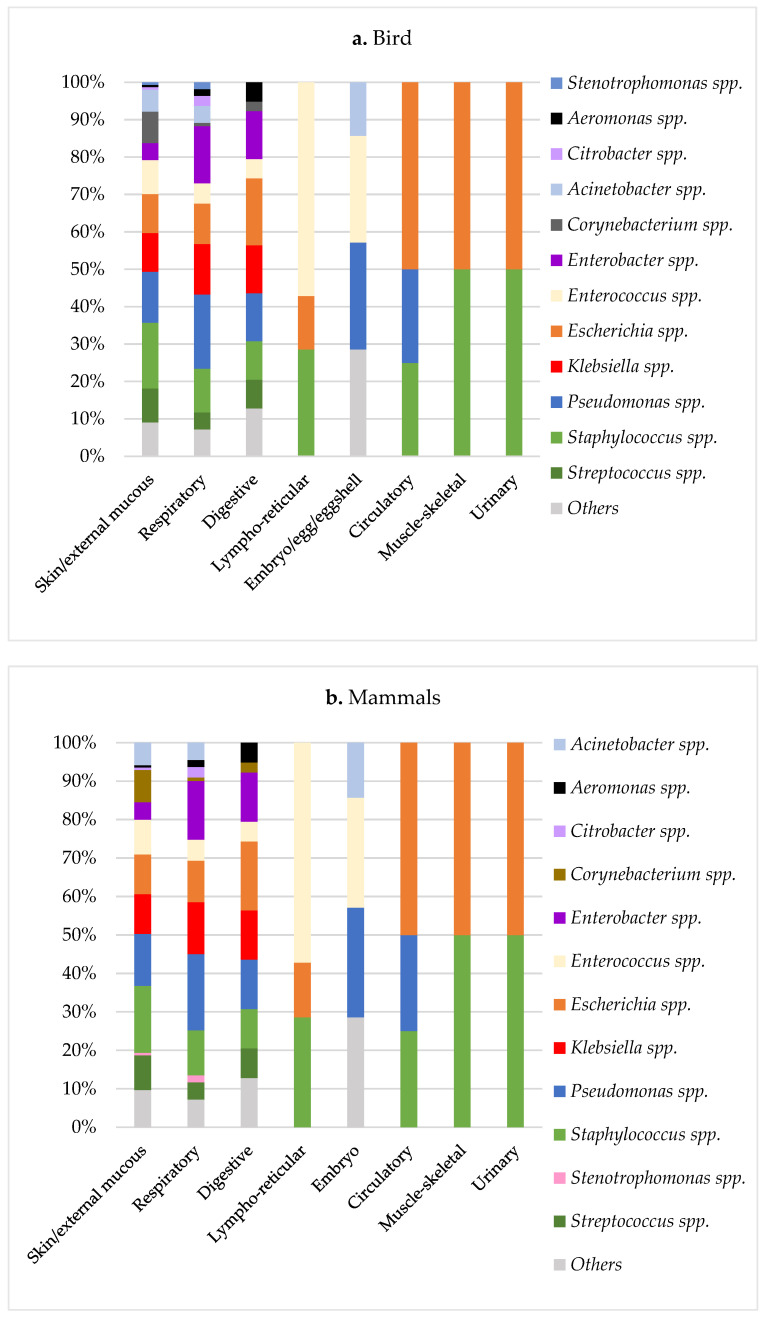
Distribution of bacterial genus according to the sample origin and the animal class: birds (**a**), mammals (**b**), and reptiles (**c**). Others include *Achromobacter* spp., *Agrobacterium* spp., *Avibacterium* spp., *Bacillus* spp., *Bordetella* spp., *Brevibacterium* spp., *Burkholderia* spp., *Chryseobacterium* spp., *Corynebacterium* spp. (in reptiles), *Cronobacter* spp., *Delftia* spp., *Elizabethkingia* spp., *Empedobacter* spp., *Haemophilus* spp., *Kocuria* spp., *Kosakonia* spp., *Leclercia* spp., *Lelliottia* spp., *Leuconostoc* spp., *Ligilactobacillus* spp., *Lysinibacillus* spp., *Mammaliicoccus* spp., *Microbacterium* spp., *Moraxella* spp., *Myroides* spp., *Neisseria* spp. (in birds and mammals), *Ochrobactrum* spp., *Pantoea* spp., *Pasteurella* spp., *Peptostreptococcus* spp., *Pluralibacter* spp., *Proteus* spp., *Providencia* spp., *Raoultella* spp., *Rothia* spp., *Salmonella* spp., *Serratia* spp., *Vagococcus* spp., *Vibrio* spp., and *Weissella* spp.

**Figure 3 animals-12-01912-f003:**
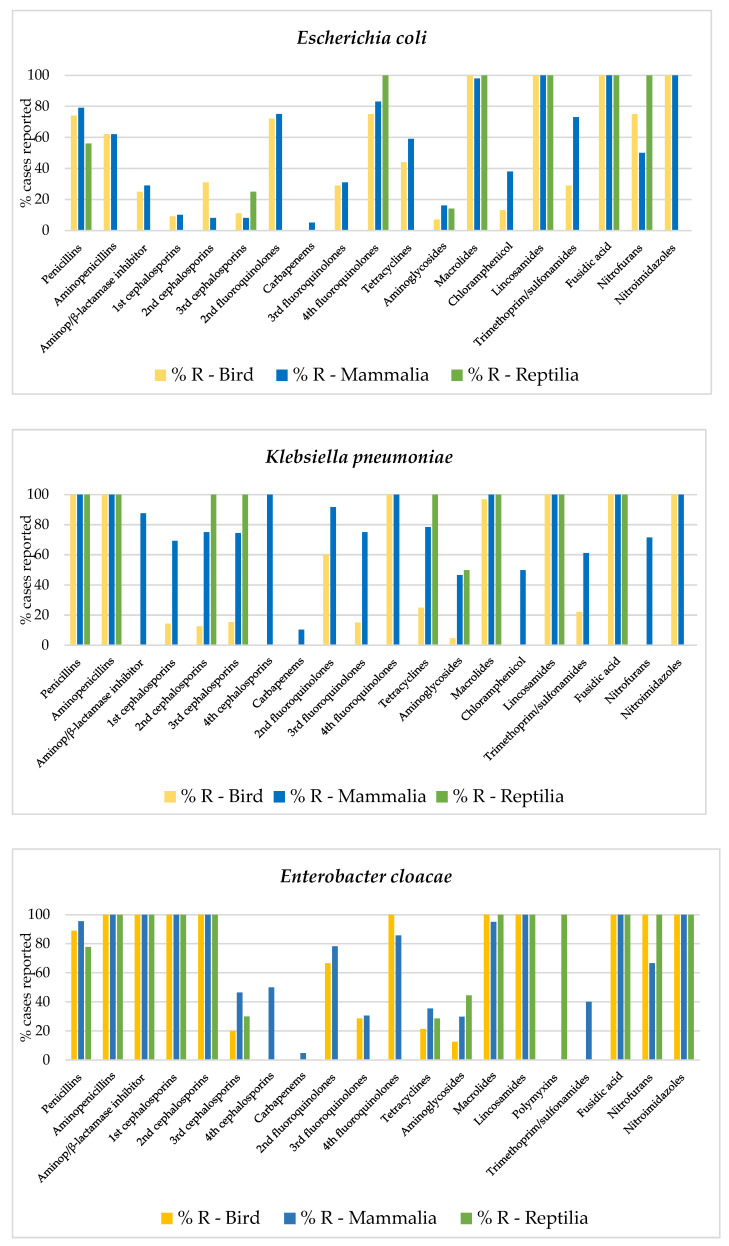
Comparison of MDR patterns (percentage of resistance) of different bacterial spp. for each animal class.

**Table 1 animals-12-01912-t001:** Number of samples according to the microbiological testing results.

Class	% Negatives (N)		% Positives (N)		Sum *
Birds	26.6	(149)	73.4	(412)	561
Mammals	18.3	(539)	81.7	(2399)	2938
Reptiles	7.8	(29)	92.2	(345)	374
Total	18.5	(717)	81.5	(3156)	3873

***** Fungi and yeasts were not considered.

**Table 2 animals-12-01912-t002:** Number and frequency of exotic pet cases analyzed by animal class and years studied from the database.

Class	2016	2017	2018	2019	2020	Total (%)
**Birds**	**53**	**73**	**62**	**121**	**103**	**412 (13)**
Columbiformes				3	1	4 (1)
Galliformes	5	7	10	15	16	53 (12.8)
Passeriformes	1	1		1	1	4 (1)
Psittaciformes	47	65	52	102	85	351 (85.2)
**Mammals**	**265**	**342**	**434**	**602**	**756**	**2399 (76)**
Carnivora	20	22	20	22	16	100 (4.1)
Eulipotyphla	2		1	1	3	7(0.3)
Lagomorpha	200	235	297	454	571	1757 (73.2)
Rodentia	43	85	116	125	166	535 (22.3)
**Reptiles**	**60**	**44**	**55**	**96**	**90**	**345 (11)**
Squamata	27	14	22	41	34	138 (40)
Testudines	33	30	33	55	56	207 (60)
**Total**	**378**	**459**	**551**	**819**	**949**	**3156 (100)**

**Table 3 animals-12-01912-t003:** Frequencies of bacterial species identified in each animal class.

	Isolations Per Animal Class (% ^1^)			
Microbiological Results	Birds	Mammals	Reptiles	Total Population
***Acinetobacter* spp. ^2^**	16 (3.9)	96 (4)	12 (3.5)	124 (3.9)
*A. baumannii*	3 (0.7)	12 (0.5)	4 (1.1)	
*A. Iwoffii*	5 (1.2)	17 (0.7)	1 (0.3)	
***Aeromonas* spp. ^2^**	5 (1.2)	18 (0.7)	34 (9.8)	57 (1.8)
*A. hydrophila*	2 (0.4)	14 (0.6)	17 (4.9)	
*A. veronii*	1 (0.2)	0	6 (1.7)	
***Bordetella* spp. ^2^**	1 (0.2)	164 (6.8)	7 (2)	172 (5.4)
*B. bronchiseptica*	1 (0.2)	155 (6.5)	5 (1.4)	
***Citrobacter* spp. ^2^**	5 (1.2)	20 (0.8)	20 (5.8)	45 (1.4)
*C. freundii*	4 (1)	10 (0.4)	14 (4)	
***Enterobacter* spp. ^2^**	35 (8.5)	152 (6.3)	14 (4)	201 (6.4)
*E. cloacae*	27 (6.5)	120 (5)	11 (3.2)	
***Enterococcus* spp. ^2^**	33 (8)	103 (4.3)	18 (5.2)	154 (4.9)
*E. faecalis*	14 (3.4)	56 (2.3)	8 (2.3)	
***Escherichia* spp. ^2^**	55 (13.3)	128 (5.3)	15 (4.3)	198 (6.3)
*E. coli*	54 (13.1)	118 (4.9)	15 (4.3)	
***Klebsiella* spp. ^2^**	46 (11.1)	146 (6.1)	16 (4.6)	208 (6.6)
*K. pneumoniae*	30 (7.3)	98 (4.1)	3 (0.9)	
*K. oxytoca*	12 (2.9)	37 (1.5)	12 (3.5)	
***Morganella* spp. ^2^**	0	6 (0.2)	14 (4)	20 (0.6)
*M. morganii*	0	5 (0.2)	14 (4)	
***Pasteurella* spp. ^2^**	2 (0.4)	198 (8.3)	1 (0.3)	201 (6.4)
*P. multocida*	2 (0.4)	144 (6)	1 (0.3)	
***Pseudomonas* spp. ^2^**	57 (13.8)	315 (13.1)	80 (23.2)	452 (14.3)
*P. aeruginosa*	44 (10.7)	217 (9)	62 (18)	
*P. fluorescens*	0	16 (0.7)	0	
***Staphylococcus* spp. ^2^**	69 (16.7)	389 (16.2)	11 (3.2)	469 (14.9)
*S. aureus*	11 (2.7)	122 (5.1)	1 (0.3)	
*S. epidermidis*	8 (1.9)	25 (1)	0	
*S. pseudintermedius*	0	20 (0.8)	0	
*S. sciuri*	7 (1.7)	9 (0.4)	1 (0.3)	
*S. xylosus*	2 (0.4)	37 (1.5)	2 (0.6)	
***Stenotrophomonas* spp. ^2^**	4 (1)	18 (0.8)	16 (4.6)	38 (1.2)
*S. maltophilia*	4 (1)	18 (0.8)	16 (4.6)	
***Streptococcus* spp. ^2^**	30 (7.3)	236 (9.8)	17 (4.9)	283 (9)
*S. intermedius*	0	26 (1.1)	0	
**Other spp.**	54 (13.1)	410 (17.1)	70 (20.3)	534 (16.9)
**Total**	412 (100)	2399 (100)	345 (100)	3156 (100)

**^1^** Percentage related to the total number of samples per animal group and the total population. **^2^** All species are included. Chi-square (X2) test applied to all the results showed no significant differences (*p*-value > 0.001).

## Data Availability

The data that support the findings of this study are limited available from the corresponding author upon reasonable request. Restrictions apply to the availability of these data, which were used under license for this study.

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
