# Peer review of "Antimicrobial Resistance in Bacteria Isolated from Exotic Pets: The Situation in the Iberian Peninsula"

_animals, 2022, doi:10.3390/ani12151912_

Round 1

Reviewer 1 Report

The present study aimed to evaluate microbiological diagnoses and the bacterial susceptibility to antibiotics in exotic pets from the Iberian Peninsula. Extensive editing of the English language and style required is required. The introduction should be improved, as well as the discussion. The extensive form should precede the use of abbreviations.

Author Response

Review Report Responses

Manuscript ID: animals-1787221

Dear reviewers,

Thank you very much for your constructive criticism of the manuscript. We have tried to give answers to all your concerns and modified the text according to your suggestions.  All modifications and revisions are registered (track changes and highlighted) in the revised version.

  1. Extensive editing of the English language and style required is required.

Answer: The text has been rechecked for English language edition

  1. The introduction should be improved, as well as the discussion.

Answer: we agreed. The introduction and discussion has been modified with track changes activated to see the modifications.

  1. The extensive form should precede the use of abbreviations.

Answer: we totally agree with the reviewer comment, and we have made the corresponding corrections.

  1. Figure 2, please make it as supplementary table.

Answer: we have move it as annex figure 1.

  1. Line 159. Which year of CLSI?

Answer: This information was already included in the text (line 130) as follows: “according to Performance Standards for Antimicrobial Susceptibility Testing for bacteria isolated from animals (M31-A3, CLSI VET01, 2008) and humans (M100-S24, CLSI, 2016) for monitoring resistant microorganisms as a potential risk to public health. “This is highlighted in yellow (line 130).

  1. Please define what was considers as MDR in these data/report?

Answer: this information was reported in lines 158-159 (in yellow).

  1. I strongly recommend to make new tables/graph showing the AMR pattern based on AWaRe classification, so that easily we know the status of resistance against A/Wa/Re groups of antibiotics. For example, what is the resistance pattern of E. coli in mammals against reserve group of antibiotics. This kind of information is very crucial and important to show here.

Answer: new tables were included in the annex with this complementary information. See annex Table 1.

  1. Line 246. Any explanation/speculation in discussion why E. coli showed intermediate levels of AMR compared to Enterobacter, Klebsiella, Citrobacter or Serratia spp.

Answer: we cannot drive any explanation but this findings are consistent with other published studies conducted by our team in other pets (cats and dogs), where enterobacteria like K.pneumoniae, Serratia marecescens or C. freundii presented also higher levels of AMR compared to E.coli strains.

  1. Line 387. Please suggest how veterinarians and physicians can work together.

Answer: The suggestion has been included in Lines 438-440: “From the One Health standpoint, it is critical that veterinarians and physicians work together to optimize, rationalize, and prudently use of antimicrobial therapies in domestic, companion, and exotic animals and humans since most bacterial pathogens and their resistance mechanisms can be shared between animals and humans. Examples include adopting interrelated education on zoonotic infections and ownership implications in health, as well as exchanging data sources, experiences, and discussing clinical cases.”

  1. Choose a different title, this one is redundant. (i.e. Relevance of antibacterial resistance in bacteria isolated from exotic pets: the Iberian peninsula reality or diffusion of antimicrobial resistance among Iberian exotic pets).

Answer: the new title is: Antimicrobial resistance in bacteria isolated from exotic pets: the situation in the Iberian Peninsula

  1. Changes in capitalized words, changed to italics, grammar inconsistencies and paraphrasing.

Answer: See, for large changes, to lines 27, 33, 43, 63, 116, 394 - 417.

  1. Lines 55-61: Attention to the citation style, it is wrongly arranged.

Answer: The citation style was checked and according to the software used seems correct.

  1. Move the title of figure 3 before the figure itself.

Answer: The article's structure specifies that the figure titles should appear in below the figures.

  1. In all the figures the title of Y axis is missing.

Answer: the figure titles have been checked.

  1. Lines 16 and 21 – change antibiotics to antimicrobials.

Answer: Agreed.

  1. Lines 70-75 are repeated in lines 76-81.

Answer: Agreed

  1. Consider transmission from humans to animals, not just the other way around.

Answer: Agreed. See line 56.

  1. Lines 104-107. The numbers presented here (and table 1 in the results) are related to all databases. However, as the authors explain in lines 183-186, after the cleaning of duplicated results, etc, the final number of samples was 3,156. “The results of the microbiological cultures were performed once the database was cleaned from duplicated results and the fungus cultures were removed. Consequently, the final data of the microbiological testing was 3,873 samples: 82% (3,156) of them presented a positive diagnosis confirmation of a pure or majoritarian bacterial infection.” In my opinion, the criteria in the “study design and animal samples” should be only the samples with a conclusive bacterial growth, with a positive diagnosis confirmation of a pure or majoritarian bacterial infection. The other samples should not be included in the study. Considering this, data in table 1 and lines 170-179 must be corrected to a total sample of 3,156 which is, in the reality, the population that can be studied with clear and unbiased results. In this perspective, table 2 of results should not exist, since doesn’t bring any useful information.

Answer:  We would like to thank the reviewer's suggestion, however, from the clinician's point of view is also important to know which percentage of microbiological cultures has no conclusive result. Since the sampled animal was suffering from a clinical infection with suspected bacterial etiology, the sample was taken and sent to the laboratory. So, all these submitted cases should be considered (regardless of the culture results), because it is useful to know the casuistic of vets that are requesting for microbiological diagnosis and antibiogram. Moreover, it is interesting to know the success of the microbiological diagnosis since a negative result can result in a second sampling collection and delivery. Considering these facts, we think that both tables are necessary for providing complementary data useful for assessing the vet practices in exotic pets.

Reviewer 2 Report

AMR is a global health crisis. Most of the studies are focused in human and domesticated animals and poultry. There is lack of adequate data on AMR in pet animals/exotic animal. These data are crucial since exotic pet could be potential sourer for AMR for human and therefore of having major public health significance. This manuscript focused on AMR in exotic pets to fill up some of the gap we have in this field.

This is a very well designed straight forward study reflecting the findings of analysis of data available on AMR in exotic pet using various analysis in Iberian Peninsula having public health impact. I have few comments as follows:

Figure 2, please make it as supplementary table, its not showing data related to AMR directly

Line 159. Which year of CLSI?? May be many years….

Please define what was considers as MDR in these data/report?

I strongly recommend to make new tables/graph showing the AMR pattern based on AWaRe classification, so that easily we know the status of resistance against A/Wa/Re groups of antibiotics. For example what is the resistance pattern of E. coli in mammals against reserve group of antibiotics. This kind of information is very crucial and important to show here.

Line 246. Any explanation/speculation in discussion why E. coli showed intermediate levels of AMR compared to Enterobacter, Klebsiella, Citrobacter or Serratia spp.

Line 387. Please suggest how veterinarians and physicians can work together

Author Response

(The authors gave the same response as above.)

Reviewer 3 Report

See the attached file. Use Adobe Acrobat Reader to see specific comments

Author Response

(The authors gave the same response as above.)

Reviewer 4 Report

The manuscript “Antimicrobial resistant bacterial infections of public health concern in exotic pets of the Iberian Peninsula” focus an important subject in equine veterinary medicine and with impact in the “one health” issue.

The abstract and introduction are clear and correct.

The methodology is acceptable, but he study design and animal samples can be improved, converting the manuscript more objective and less misleading.

The results section can be improved after correction in the methodology.

The discussion and conclusions are correct. However, regarding the “one health concept”, the hypothesis that some of the MDR observed in the study could also be transmitted from humans to animals should be considered. Not only on the other way.

Simple summary

Lines 16 and 21 – change antibiotics to antimicrobials.

Introduction

Lines 70-75 are repeated in lines 76-81.

Materials and methods + results sections

Lines 104-107. The numbers presented here (and table 1 in the results) are related to all database. However, as the authors explain in lines 183-186, after the cleaning of duplicated results, etc, the final number of samples was 3,156.

“The results of the microbiological cultures were performed once the database was cleaned from duplicated results and the fungus cultures were removed. Consequently, the final data of the microbiological testing was 3,873 samples: 82% (3,156) of them presented a positive diagnosis confirmation of a pure or majoritarian bacterial infection.”

In my opinion, the criteria in the “study design and animal samples” should be only the samples with a conclusive bacterial growth, with a positive diagnosis confirmation of a pure or majoritarian bacterial infection. The other samples should not be included in the study. Considering this, data in table 1 and lines 170-179 must be corrected to a total sample of 3,156 which is, in the reality, the population that can be studied with clear and unbiased results. In this perspective, table 2 of results should not exist, since doesn’t bring any useful information.

Author Response

(The authors gave the same response as above.)

Round 2

Reviewer 1 Report

well done. Thank you

Author Response

No other comments. Thank you

Reviewer 3 Report

Dear Authors, 

Thank you for the revised version of your manuscript

Author Response

Than you very much for your positive feedback and your constructive suggestions.

Reviewer 4 Report

The manuscript has improved.

The authors tried to follow all the suggestions.

However, my main concerns have not been corrected.

In line 30, the total sample is 3,873. The same in line 177.

Then in line 32 is 3,156. The same in line 177.

Finally, in line 96, the total sample is 3,986.

This is confusing and misleading to the readers.

Its not an error, but could be improved.

By the other hand, there can be lots of causes why a microbiological culture has no conclusive results: errors in sampling, transportation, previous treatments, etc.

So, the explanation to include the no conclusive samples in the results has no value to the clinician’s point of view. So, I do not agree with the authors.

I keep my previous suggestions:

The criteria in the “study design and animal samples” should be only the samples with a conclusive bacterial growth, with a positive diagnosis confirmation of a pure or majoritarian bacterial infection. The other samples should not be included in the study.

Data in table 1 and lines must be corrected to a total sample of 3,156 which is, in the reality, the population that can be studied with clear and unbiased results. In this perspective, table 2 of results should not exist, since doesn’t bring any useful information.

Author Response

We have followed the reviewer's suggestions and changed the results accordingly. In the R2 version the results are showing as follows:

- Lines 165-168: The results of the microbiological cultures were performed once the database was cleaned from duplicated results and the fungus cultures were removed. Consequently, the final data of the microbiological testing was 3,156 samples which presented a positive diagnosis confirmation of a pure or majoritarian bacterial infection (Table 1).

 -Lines 172-73: From the microbiological results, birds (n=412) represented 13%, mammals (n=2,399) represented 76% and reptiles (n=345) 11% of the total sample (Table 2). 

-Table 2: 

Table 2. Number and frequency of exotic pet cases analyzed by animal class and years studied from the database.

Class

2016

2017

2018

2019

2020

Total (%)

Bird

53

73

62

121

103

412 (13)

Columbiformes

3

1

4 (1)

Galliformes

5

7

10

15

16

53 (12.8)

Passeriformes

1

1

1

1

4 (1)

Psittaciformes

47

65

52

102

85

351 (85.2)

Mammalia

265

342

434

602

756

2,399 (76)

Carnivora

20

22

20

22

16

100 (4.1)

Eulipotyphla

2

1

1

3

7(0.3)

Lagomorpha

200

235

297

454

571

1,757 (73.2)

Rodentia

43

85

116

125

166

535 (22.3)

Reptilia

60

44

55

96

90

345 (11)

Squamata

27

14

22

41

34

138 (40)

Testudines

33

30

33

55

56

207 (60)

Total

378

459

551

819

949

3,156 (100)

We hope that changes can be accepted by the referee.

Thank you very much for your constructive criticism.